# How School Travel Affects Children’s Psychological Well-Being and Academic Achievement in China

**DOI:** 10.3390/ijerph192113881

**Published:** 2022-10-25

**Authors:** Pengxiang Ding, Suwei Feng

**Affiliations:** School of Public Economics and Administration, Shanghai University of Finance and Economics, Shanghai 200433, China

**Keywords:** school travel, commuting time, commuting mode, children, psychological well-being, academic achievement, China

## Abstract

Previous research on the role of school travel in children’s well-being (WB) has paid little attention to developing countries. Using national survey data across China, this study examines how children’s psychological well-being (PWB) and academic performance differ across commute duration and mode among urban, rural, and urban fringe areas. Our findings show that commute times are significantly negatively associated with children’s PWB and academic achievements, and this correlation varies across areas. Children living in the urban fringe have the longest average one-way commuting time (18.6 min), but they have a better acceptance of longer commuting duration, whereas commuting time is more influential in the city center and rural areas. Regarding travel mode, walking to school is positively associated with PWB in the center area, while bicycles and public transport positively affect the rural student scores. Results from quantile regression show that students on the lower quantiles of the conditional distribution of PWB tend to suffer more than the others when commuting time increases; students with middle scores respond similarly to marginal changes in commuting time. Recommendations for urban planners and policymakers to enhance child WB include fostering school-home balance, improving public transit services, and investing in pedestrian and bicycle infrastructure for those vulnerable groups.

## 1. Introduction

The latest OECD Programme for International Student Assessment (PISA2018) report shows that Chinese secondary school students ranked first in reading, mathematics, and science among 79 countries and regions. However, results from the well-being assessments in PISA2018 show that Chinese students have the lowest scores (ranked 61st) for life satisfaction and require comprehensive interventions. It is widely known that reducing commuting time is an efficient intervention to increase positive emotion as commuting time is found to have a negative effect on adult psychological well-being in most research [1,2,3,4,5,6]. What remains unclear is to what extent the psychological well-being of school-age children also suffers from long commute times. 

In addition, previous studies in various countries have found long commute duration negatively affects primary and secondary school students’ academic performance [7,8,9], but whether the relationships between commuting and students’ academic performance differ for different groups (e.g., between urban and rural areas) is still unclear. Land use and public service facilities in the urban fringe and rural areas are significantly different from the urban center in China. The low density and the distance between homes and schools make it more difficult for rural residents to reach their destinations [10,11]. Thus, school commutes may negatively affect students’ academic performance in rural areas. Chinese secondary school students’ achievements contain urban–rural disparity. The average PISA2018 score of urban and rural students differs by 28 points [12]. What remains unclear is whether transport may have widened the achievement gap between rural and urban students. Understanding the links between children’s school travel behavior and academic achievement may provide important suggestions for policymakers on how children’s academic achievement in different areas can be improved. 

There is a growing body of research examining the relationship between school travel and WB reviewed by the authors of [13,14,15,16,17] and, most recently, Phansikar et al. (2019). However, existing studies on this relationship are mostly from developed countries [17,18,19]. To the authors’ knowledge, as the world’s largest school system, only three empirical studies on school-aged child WB are from China [20,21,22]. Of the three studies, only Sun et al. (2015) specifically examined how commute characteristics, such as commute mode, were associated with children health outcomes [20].

Empirical evidence in previous literature on child well-being in the western context may not apply directly to China’s unique situation because there is a big difference between developed and developing countries. First, China implements the nearby enrollment policy in which students enter the school according to their residence. However, the insufficient supply of high-quality schools leads to much higher housing prices in better school districts [23]. Some parents can’t afford school district housing and instead buy old or small housing units in school districts but do not live there. This spatial separation of residential homes and enrolled schools increases school-to-home distance [24,25]. According to a survey, 29.3% of primary school students live 10 km or farther from their schools in Beijing [26]. Second, the policy of school merging was implemented in rural areas from 2000 to 2010. The commuting distance of rural students was extended from 1.6 km to 4.0 km after the layout adjustment [27]. Third, active commuting to school has also decreased in China as the number of motor vehicles increases. About 84% of Chinese children were actively commuting to school in 1997, and the rate decreased to 55.8% by 2010 [20,28]. Nowadays, parents are more inclined to drive their children to school, accounting for 15% of the morning peak traffic in some larger cities [24]. Given these unique contexts, it is important to conduct empirical studies of children’s WB and examine how well-being differs across various modes and by commute duration in Chinese urban and rural areas.

We identify four gaps in the literature that motivate this research. First, previous research on child commute well-being has been dominated by studies from the U.S., Canada, and Europe, while few studies have paid specific attention to children in developing countries. Second, a significant correlation between commuting and SWB has been mostly based on surveys of a single city [21,22,29]. The conclusions from one city are difficult to generalize to other cities. Third, there are significant differences in the commuter patterns of different groups (e.g., those living in different areas), and it is unclear how much of a gap there is in mental endurance of commuting duration among the children in the urban, rural, and urban fringe areas. Finally, very few papers crossed well-being domain boundaries in relation to commuting, and the cognitive domain is less well studied. Previous studies that measure the cognitive domain have focused on cognitive ability, while students’ academic performance may be more critical for long-term development and are rarely studied [30].

The aim of this study is to examine the relationships between commuting, children’s psychological well-being, and academic performance using the Chinese Education Panel Survey (CEPS2014–2015) data. In order to fill the aforementioned knowledge gaps, this study identifies: (1) How commuting (time/mode) influences a child’s psychological well-being; (2) How commuting (time/mode) influences child’s academic performance; and (3) Whether commute time, child’s psychological well-being, and academic performance differ across different groups (considering urban and rural areas, psychological WB and academic performance distribution).

The rest of this paper is organized as follows. Section 2 provides a literature review on the relationship between commuting and child WB. Section 3 introduces the data and the modelling approach. Section 4 provides descriptive statistics of the variables and estimation results. Section 5 presents the important findings. Section 6 concludes.

## 2. Literature Review

### 2.1. Children’s Well-Being

Well-being (WB) is a term that is commonly used but difficult to define [31]. A number of studies have been conducted on WB from two different perspectives. First, according to the Hedonic perspective, WB is related to happiness, life satisfaction, and quality of life [32,33].Second, the eudaimonic perspective focuses on people’s self-actualization and personal growth [34,35]. No matter which perspective is adopted, there seems to be a consensus that WB is a multidimensional indicator [31,36,37]. 

Similarly, children’s well-being, measuring the condition of their lives, cannot also be represented by a single indicator [38,39]. According to Pollard and Lee (2003), children’s well-being can be defined as a multidimensional concept across five domains: physical, psychological, social, cognitive, and economic [37]. Furthermore, they distinguished between psychological well-being and cognitive well-being. The term psychological well-being refers to indicators of emotional well-being, mental health, and illness, while cognitive well-being refers to indicators of intelligence and school performance (such as academic performance). A recent integrative review by Waygood et al. (2017) examines how transport impacts on children’s five domains of WB, two of which relate to psychological and cognitive domains [17,40]. In the following sections, we will discuss these two types of WB in more detail.

### 2.2. Commuting and Psychological Well-Being

Psychological well-being (PWB) has been defined as a “state of well-being in which an individual realizes his or her abilities, can cope with the normal stresses of life, can work productively, and can make a contribution to his or her community” (WHO 2018). This subjective version of WB has two dimensions: (1) Hedonic well-being, which refers to a positive/negative emotional mood, and (2) Eudaimonic well-being, which representing a high purpose such as finding meaning, personal growth, and self-fulfillment [17,33]. This study focuses on the hedonic PWB.

Many studies have explored the relationship between commuting time and psychological well-being [41,42,43]. It is believed that reducing commuting time is an efficient method to increase positive emotion because commuting time is found to have a negative effect on residents’ PWB in most research [1,2,3,4]. For instance, Nie and Sousa (2018) found that Chinese residents with longer commuting times tended to have lower levels of happiness [3], and similar findings on this relationship were obtained by Sun et al. (2021) and Zhu et al. (2019) [5,6]. Other studies, however, found no significant relationship between commuting time and PWB [44,45,46,47]. Thus, it is still debatable whether commuting time significantly influences adult psychological well-being. A relatively small number of studies found a negative association between PWB and commute time for children and adolescents [19,47]. For example, Westman et al. (2017) found that a short journey (less than 15 min) for children resulted in a more positive mood at school [19].

Besides commuting time, increasing numbers of studies are paying attention to the impacts of commuting mode on child PWB. For example, a study using children’s travel data from Quebec City, Waygood and Cervesato (2017) found that children who bicycled to destinations had the highest happiness levels [48]. Similarly, using data from primary and secondary schools in the City of Vienna, Juliane Stark et al. (2018) found a positive association between active travel and children’s PWB [49]. Based on a study of non-westernized contexts, Leung and Loo (2017) focus on primary school children in Hong Kong. Children engaging in active transport rate their journeys as happier than those using motorized transport [21]; Sun et al. (2015) studied the transport mode to school among children in China, and found that more active commuting to school might help to decrease the risk of obesity and depressive symptoms [20]. Generally, there are consistent findings that positive experiences while active commuting [17].

### 2.3. Commuting and Academic Achievement

The cognitive domain refers to intellectual or school-related measures, including cognitive abilities, academic achievement, and concentration [37]. Using the survey data of grade 4, 6, and 8 students in Vimoran County, Sweden, Westman (2017) found that commuting time greater than 15 min had a positive impact on cognitive ability. The possible mechanism is that students can use smartphones to study or communicate with other activities during commuting [19]. Westman (2020) further defines the differences between cognitive abilities and academic achievement: “academic achievement is related to children’s school grades, while cognitive ability is a broader term”, and academic achievement is much less in focus than cognitive abilities in prior studies [30]. Therefore, this study also focuses on student academic achievement measured by mid-term exam scores collected from 112 schools in China.

Research on commuting and academic performance has been increasing in recent years, but existing studies have not reached a consensus. Long commute duration may negatively affect primary and secondary school students’ attendance [50,51]. School absenteeism reduces academic performance (verbal and mathematical scores) in the short term and will hurt students’ future educational decisions in the long run [9]. In addition, long commuting time will increase students’ mental stress and negative emotions, making it difficult to concentrate in class and have less time for exercising and sleeping [52,53,54]. Tigre et al. (2017) surveyed 2643 sixth graders child in Brazil, using econometric approaches based on causal inference, and verified the negative relationship between commuting time and students’ grades [55]. However, Contrerans et al. (2018) studied the relationship between commuting time and academic performance of 8th-grade students in Santiago, Chile, and found that the coefficients were not always significant [56].

Moreover, there was no consistent conclusion between commute mode and academic performance. Several studies have found the positive effects of active travel on children’s cognitive abilities, such as spatial knowledge [52,53]. Studies from Switzerland also showed that students who rode bikes to school were more active, and learned more in class [54]. However, Dijk et al. (2014) found no significant relationship between active commuting and students’ cognitive performance in Dutch adolescents. However, girls were found to perform better on attention tests, and gender might have moderated the effect [55]. 

In sum, the relevant research on children’s school travel is still in the initial stage. This study aims to fill the research gap in school travel and child crossed-WB relations in a developing country context. First, this study examines how commuting (time and mode) influences child’s PWB and academic performance, which will help us to better understand the various impact of commuting on the well-being of children in China. Further, this study compares the commuting, PWB, and academic performance relationship among urban, rural, and urban fringe areas, which will improve our understanding of differences in the effects of different groups in China’s context.

## 3. Methods

### 3.1. Data

The data for this paper are drawn from the China Education Panel Survey (CEPS), the first nationally representative longitudinal survey of junior-high students conducted by the National Survey Research Center at the Renmin University of China. The survey includes extensive information on students’ socio-demographics, travel mode, test scores, school management, and teacher qualities. The baseline survey is a random sample (applied a multistage sampling method with probabilities proportional to size) of approximately 20,000 students in 438 classrooms of 112 schools in 29 county-level units in mainland China in 2013–14 and 2014–15. Only 2014–15 CEPS first collect student school travel mode and commute time; therefore, we use a wave of 2014–2015 CEPS. After excluding the sample of boarding students, the final sample includes 4807 8th-grade students.

### 3.2. Models and Variables

The regression model used for the analysis is below:$$WB_{sji}\;=\beta_{0}+\beta_{1}\;CT_{i}+\beta_{2}\;Mode_{i}+\beta_{3}\;Child_{i}+\beta_{4}\;Family_{i}+\beta_{5}\;School_{s}+\gamma_{j}+\varepsilon_{sji}$$
where $WB_{sji}$ denotes the well-being of child *i* in terms of PWB or academic achievement, and $CT_{i\;}$ denotes home-school commute time (one way) of child *i*. $Child_{i}$ is a vector of individual *i*’s characteristics, and $Family_{i}\;$ is a vector of family characteristics. $School_{s}$ is a vector of school characteristics, $\beta_{1}$ and $\beta_{2}$ are the key coefficients of interest, and $\varepsilon_{sji}$ is the error term.

The dependent variables are PWB and academic achievement. PWB is measured by the respondents’ reports on how often, in the last 1 week, they felt: (1) Depression; (2) Too depressed to concentrate; (3) Unhappy; (4) Life is boring; (5) Can’t work hard; (6) Sadness; (7) Tension; (8) Worry; (9) Have a hunch that something bad will happen; (10) Too energetic and inattentive in class. Respondents were asked to rate their PWB on a five-point Likert-type scale ranging from (1) never to (5) always. We reversed the ten emotional indicators and used the 0–1 standardized method to construct a PWB index variable with a value range of 0–100. The 2014–2015 CEPS records mid-term test scores in math, English, Chinese, and academic performance are measured by the mean of these subjects’ scores.

The independent variables are daily commute time (one-way) and mode. In the 2014–2015 CEPS, students were asked about their transportation mode and time spent. The question is: “*What mode do you usually use from home to school?*”, “*How long does it usually take you from home to school (one-way) using the transportation mode above? (minutes)*”. 

The control variables in this study are roughly divided into three sets: (1) Individual characteristics: age, gender, only-child (whether a student is the only child in the family), and hukou. (2) Family background information: parents’ education experience, family financial situation, and parents’ expectations. (3) School information: School ranking, teacher mental health training, teacher and student ratio, and teacher quality. We also employ county-level fixed effects to control for the unobserved contextual factors.

## 4. Results

### 4.1. Descriptive Statistics

As shown in Table 1, the mean (one-way) commute time of the respondents in the sample is 17.56 min. Children preferred to walk (41%) or bicycle (21%) on their school trips, which is consistent with existing studies in Western countries, as well as recent studies in China [20,57]. Significant differences exist in commute time and mode among urban, rural, and urban fringe areas. Respondents who live in the urban fringe have the longest average commuting time, which is 18.6 min. The proportion of active commute in rural areas is higher than that in urban and urban fringe areas, probably because some children cannot afford the transportation expenses and commute by foot or bicycle. There is little difference in the proportion of walking and bicycle trips between urban and fringe areas. The proportion of public transport use in urban and fringe reaches 19 % and 21%, which is higher than that of rural areas (12%). This result partly reflects differences in the availability of public transport services in the different areas. The proportion of cars traveling from the urban center to the rural areas shows a downward trend. Motor vehicle use in rural areas is relatively higher than in urban and urban fringe areas, which is consistent with western countries [58].

As shown in Table 2, the children’s PWB (70.71) in the urban centers is the highest, and the mid-term test scores of urban students reached 69.38, which is also higher than those of the periphery (66.72) and rural areas (60.88). Child individual characteristics, family background information, school, and class level in different areas show great differences. The proportion of only-child in urban areas is the highest, and parents’ education experience, families’ economic status, and schools are better than those in urban fringe and rural areas.

### 4.2. The Impact of Commuting on PWB

Table 3 shows the regression results. We find a negative association between time spent traveling to school and PWB (model 1). Every extra 10 min (one way) reduces PWB by 1.15 points on the 100-point scale. Boys, only-children, and high-income families are more likely to report higher PWB, and school rank shows a negative association with child PWB. 

Moreover, we found a correlation between commute mode and PWB. Using other modes (car/motor) as a reference group, walking to school is associated with a higher PWB (model 2) for the student in the urban centers. There are no such significant correlations in urban fringe and rural areas.

With respect to differences in the area (Table 3 models 2–4), we find that the commute duration is significantly negatively correlated with child PWB in all three areas. It is interesting to note that students living in urban centers are most affected (every extra 10 min reduces their PWB by 1.32 points). 

**Table 3 ijerph-19-13881-t003:** Estimation results of the influences of explanatory variables on child well-being.

	Psychological				Performance			
	All	Center	Periphery	Rural	All	Center	Periphery	Rural
	Model 1	Model 2	Model 3	Model 4	Model 5	Model 6	Model 7	Model 8
**Commute time**	−0.115 ***	−0.132 ***	−0.130 *	−0.113 ***	−0.077 ***	−0.069 **	−0.014	−0.095 ***
**Commute mode**								
Walking	−0.049	2.435 *	−2.749	−0.178	−0.613	−1.002	−0.768	1.885
Bicycle	1.537	1.527	1.848	2.158	0.729	−0.216	0.143	2.453 *
Public transit	0.150	2.329	−1.955	0.088	0.685	0.063	−1.456	5.296 ***
**Individual**								
Age	−0.627	0.172	−1.190	−1.077	−2.007 ***	−1.464 ***	−1.888 ***	−2.140 ***
Gender	1.922 ***	2.543 ***	1.644	1.255	−6.593 ***	−5.429 ***	−7.893 ***	−6.434 ***
Only-child	1.157	0.497	1.767	1.130	1.018 **	1.183	−0.075	1.155
Hukou	0.849	0.965	2.687*	−2.099	2.274 ***	3.241 ***	0.704	1.823
**Family**								
Father education	0.335	0.227	−0.681	2.498	2.171 ***	2.330 ***	2.139 **	1.043
Mother education	0.133	2.360 **	−1.631	−3.123 *	1.517 ***	1.415 *	2.032 **	1.122
Parent expectation	2.359 ***	1.377	3.603 **	2.572 **	13.829 ***	12.730 ***	13.953 ***	14.493 ***
Family financial	2.496 ***	1.992*	3.142 **	2.570 ***	0.565	0.526	−0.028	0.344
**School**								
Teacher student ratio	−0.119	−0.328 *	0.167	−0.214	−0.054	0.064	−0.424 ***	−0.326 ***
Teacher mental health train	1.031	0.347	0.681	1.098	0.228	−0.529	1.342	−0.090
Teacher quality	0.005	−0.007	0.019	0.008	0.071 ***	0.072 ***	0.028	0.141 ***
School rank	−1.216 **	0.857	−2.496 **	−1.668	3.528 ***	4.025 ***	5.544 ***	0.205
Constant	73.815 ***	58.588 ***	82.303 ***	84.417 ***	73.437 ***	63.680 ***	71.617 ***	80.627 ***
Observations	4807	2135	1269	1403	4807	2135	1269	1403
R-squared	0.040	0.049	0.064	0.064	0.448	0.465	0.439	0.440

Note: * *p* < 0.1; ** *p* < 0.05; *** *p* < 0.01.

### 4.3. The Impact of Commuting on Academic Achievement

As reported in Table 3, the results indicate that when commute time increases, child academic achievement reduces, and every extra 10 min (one way) reduces middle exam scores by 0.77 points (model 5). Commute mode is not associated with academic achievement. Among the controlled variables, urban hukou, only-child, girl, high-income family, and parents’ expectations have positive correlation with child’s performance. We find that a highly ranked school can improve students’ academic performance and reduce their PWB. The reasons may be that students in highly ranked schools get good educational resources, but also face fierce competitive pressure.

Using bicycles and public transportation increases children’s academic performance in rural areas, but does not have a significant effect in urban and fringe areas. A possible explanation for this is that in China’s rural areas, transportation improvement is a top priority in poverty alleviation strategies, and public transportation services have been improved [59].

Table 3 also presents the results of the relationships between commuting time and residents’ academic achievement in different areas. The result shows that commuting duration is related to child academic achievement in the center and rural areas. In contrast, commute time in urban fringe areas shows no significant correlation with child performance. This finding indicates that children living in urban fringe areas may better accept a longer commuting duration. Due to the spatial separation between residential homes and enrolled schools, children’s school-to-home distance increases, but this might have been compensated by a better school, so school rank has a particularly strong influence on students living on the urban fringe. 

### 4.4. Relationship among Different Commuting Time Groups and Child WB

For a deeper understanding of the effect of different commuting time groups on child well-being, commute duration is defined as five dummy variables in Table 4. We find that 20 ≤ commuting time < 30 and 30 ≤ commuting time < 40 and 40 ≤ commuting time show negative relationships with child WB when compared to the reference group (commuting time < 10 min). Further, the result also shows that the coefficients increase along with commute duration, indicating that longer commute duration is more strongly related to PWB and academic achievement. 

We found that the relationship between commuting mode and well-being remained consistent with the results in Table 3. Walking to school in urban centers brings students more positive emotions compared to other commuting modes, and no such effect was found in other areas. Meanwhile, bicycle and public transportation commuting in rural areas had a positive effect on students’ academic performance.

With respect to different areas, students living in the urban center are most vulnerable to suffering from commute duration; commuting time ≥ 20 would have negative effects on their wellbeing. In rural areas, commuting time ≥ 30 negatively affects children’s PWB and academic performance than the benchmark commute time. More than 40 min of commuting in the periphery areas have a negative effect on PWB and no effect on performance. This finding is consistent with the result of the basic model in Table 3.

### 4.5. Quantile Regression

This section investigates if students performing on the upper quantiles of the conditional PWB and grade distribution respond differently to commuting duration when compared to those performing on the bottom quantiles of the distribution. We analyze heterogeneity via a quantile regression model. The graph in Figure 1 shows coefficients estimated for every decile of the conditional WB distribution. Commuting time has a negative impact on students with different PWB and academic performance. On the left-hand side of Figure 1, we observe larger effects on students located on the lower quantiles of the conditional distribution of PWB. Students who report bad PWB tend to suffer more than others when commuting time increases. The graph on the right-hand side of Figure 1 presents the coefficients estimated for student scores. Interestingly, students with middle scores are quite uniform along with the distribution.

## 5. Discussion

Based on nationally representative CEPS data, we find that commuting time and mode both affected children’s PWB and academic performance in the urban, urban fringe, and rural areas in China. 

First, we find that long commuting time is associated with lower PWB in China. It is consistent with results from a study involving Swedish children [19], which found that a short journey results in a more positive mood. Janáček (2020) also observed that Czech students with long commute times reported a lower level of life satisfaction, indicating that commute time plays an important role in improving student well-being [47]. Looking at differences between urban and rural areas, students living in urban centers are more sensitive to longer commute times than other areas. One reason may be that with rapidly increasing motorization in China, traffic congestion is getting more serious. People living in the city centers face more traffic and noise, and those traffic-induced noise pollution may attenuate the residents’ mental health [60]. According to a recent national study of mental health, with nearly 30 million Chinese adolescents under 17 years old suffering from mental anxiety requiring comprehensive interventions [61], our study proves that reducing commuting time is an effective intervention to enhance Chinese adolescents’ PWB.

Second, we find that commute time negatively affects students’ academic performance. This finding is in line with the evidence from Brazil and Chile [54,55], with children who traveled for longer school journeys performing significantly worse than children with shorter school journeys. Further, compared with urban centers and rural areas, there is no significant correlation between commuting time and academic performance in the urban fringe. This finding implies that there are unobserved compensatory factors associated with longer commutes, such as a better school, but child PWB is still significantly negatively affected. Thus, the spatial mismatch and its negative impacts caused by rising school district housing prices need to receive much attention in planning strategies. The negative effect of long commute time on student scores in rural areas is higher than those in the urban and urban fringe areas, indicating that commute duration may have widened the achievement gap between rural and urban students. It is necessary to narrow the gap between urban and rural areas by improving transportation facilities and providing transportation services to reduce child commuting time in rural areas.

Third, we find that walking to school positively affects children’s PWB in urban centers. In line with other studies [14,57,62,63,64], we also found that active commuting improved adolescent PWB. However, this positive effect was not significant in rural areas, possibly because the environment in fringe and rural areas is not so friendly to pedestrians and cyclists, and the stress risks caused by a bad travel environment, as well as air pollution, offset the potential positive effect of active commuting [11]. Moreover, we found bicycle and public transport have a positive effect on improving the performance of rural students. This is similar to the previous study by Muralidharan and Prakash (2017) in rural India [65], who found that providing bicycles to students increased their enrollment and the likelihood that students would pass the school certificate exam. There should be adequate public service facilities in rural areas in planning management.

Importantly, we find that 20 ≤ commuting time < 30, 30 ≤ commuting time < 40, and 40 ≤ commuting time show negative relationships with child PWB and performance when compared to the reference group (commuting time < 10 min) and the coefficients increase along with commute duration. The results indicate that commutes less than 20 min have little effect on student outcomes—either PWB or performance. The effects of commuting time on child WB vary across areas. Students tend to have better endurance in periphery areas for longer commuting times (40 min). In contrast, children in urban centers are more sensitive to commuting time (20 min). This finding further confirms the above conclusion and helps urban planners understand the different endurance by which commuting time affects happiness across the urban structure. 

Finally, results from quantile regression show variation within the quantiles of the conditional PWB and score distribution. Students on the lower quantiles of the conditional distribution of PWB tend to suffer more than others when commuting time increases. For performance, students with middle scores are quite uniform along the distribution. Those students respond similarly to marginal changes in commuting time to school, indicating that better school travel can promote performance not only for those less well-off.

If children’s PWB and performance can be influenced by school travel, it is important for policymakers to learn about this relationship to motivate and enhances children’s well-being. Some students must endure a long commute to attend higher-quality schools due to the uneven distribution of education resources. The government should find solutions to this “schools-homes” spatial separation problem, such as relocating good schools in fringe areas. Further, some students in rural areas lack transportation tools and can only go to school on foot. Thus, the government should subsidize bicycles or improve public transportation in rural areas. From the perspective of urban planning, the built environment should be better optimized, especially the street and community environment conducive to students’ active transport. 

This study has some limitations. First, we could only obtain relevant data up to the 2014–2015 wave, and changes in school travel (commuting time and mode) may have occurred since then. We will continue to advance the research as new rounds of survey data become publicly available in the future. Further, due to data limitations, this study cannot cover the detail of students’ commuting variables. For example, regarding commuting mode, we cannot observe whether there is a change in transportation mode, all of which may influence children’s well-being. Second, a cross-sectional analysis cannot be used to determine the causal-effect relationship between commuting time and child WB, and longitudinal study studies are desired to infer causal results. Finally, this study focused on the relationship between commuting, children’s PWB and achievement performance. Other domains such as social and economic aspects still need further discussion.

## 6. Conclusions

Using national survey data across China, this study examines how child psychological well-being and academic performance differ across commute duration and mode among urban, rural, and urban fringe areas. Our findings show that commute times are significantly negatively associated with children’s PWB and academic achievements, and this correlation varies across areas. Children living in the urban fringe have the longest average one-way commuting time (18.6 min), but they have a better acceptance of longer commuting duration, whereas commuting time is more influential in the city center and rural areas. Regarding travel mode, walking to school is positively associated with PWB in the center area, while bicycles and public transport positively affect the rural student scores. Results from quantile regression show that students on the lower quantiles of the conditional distribution of PWB tend to suffer more than others when commuting time increases; students with middle scores respond similarly to marginal changes in commuting time to school. Recommendations for urban planners and policymakers to enhance child WB include fostering school-home balance, improving public transit services, and investing in pedestrian and bicycle infrastructure for those vulnerable groups.

## Figures and Tables

**Figure 1 ijerph-19-13881-f001:**
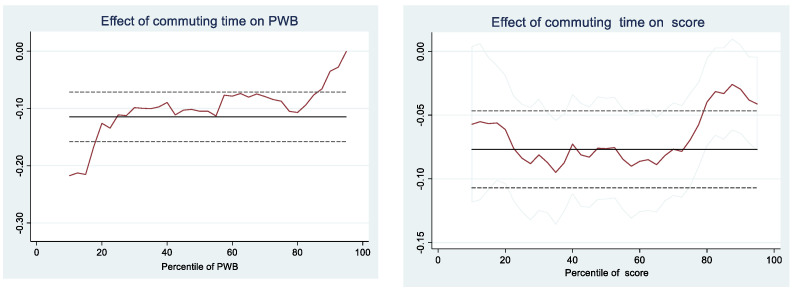
Censored Quantile Estimates.

**Table 1 ijerph-19-13881-t001:** Variable definitions and descriptive statistics.

	Measure and Value	All	Center	Periphery	Rural
**Commuting time**	Min/day, single way	17.56	16.91	18.60	17.62
**Commuting mode**					
	Walk = 1, otherwise = 0	0.41	0.41	0.36	0.46
	Bicycle = 1, otherwise = 0	0.21	0.19	0.23	0.23
	Public transit = 1, otherwise = 0	0.13	0.19	0.21	0.12
	Car = 1, otherwise = 0	0.17	0.17	0.13	0.08
	Motor = 1, otherwise = 0	0.07	0.05	0.07	0.11
**Child WB**					
Psychological WB	Scale from 1 to 100	70.35	70.71	70.47	69.69
Academic achievement	Score scale from 1 to 100	66.20	69.38	66.72	60.88
**Individual**					
Age	The age of the respondent	14.04	13.97	13.97	14.23
Gender	Male = 1	0.50	0.49	0.50	0.53
Only-child	Only-child = 1	0.54	0.66	0.56	0.34
Hukou	Urban = 1	0.36	0.50	0.38	0.14
**Family**					
Father education	Junior and above = 1,	0.43	0.60	0.40	0.21
Mother education	Junior and above = 1	0.38	0.54	0.35	0.15
Parent expectation	High = 1	0.79	0.81	0.80	0.76
Family financial	Middle and High income = 1	2.88	3.01	2.85	2.72
**School**					
Teacher-student ratio	Teacher /student	13.09	13.63	13.29	12.10
Teacher mental health train	Yes = 1, no = 0	1.13	1.16	1.15	1.06
Teacher quality	The number of the teacher has obtained a bachelor’s degree,	83.58	97.32	84.50	61.83
School rank	Scale from 1 to 5	4.01	4.23	3.95	3.73

**Table 2 ijerph-19-13881-t002:** Respondents’ well-being by a different commute modes.

	Psychological				Performance			
	All	Center	Periphery	Rural	All	Center	Periphery	Rural
Walking	70.49	72.21	69.68	68.78	63.77	66.82	65.31	58.61
bicycle	70.80	70.02	71.40	71.25	66.44	69.59	67.08	61.90
motor	70.10	67.95	71.39	70.78	68.06	72.81	68.24	64.82
public	69.63	69.84	69.68	69.02	65.00	67.13	63.93	61.57
car	70.25	69.66	71.80	69.93	74.04	76.89	73.88	64.70

**Table 4 ijerph-19-13881-t004:** Estimated results under different commuting groups.

	Psychological				Performance			
Variables	All	Center	Periphery	Rural	All	Center	Periphery	Rural
	Model 9	Model 10	Model 11	Model 12	Model 13	Model 14	Model 15	Model 16
**Commuting time (reference: 0 <** **commuting time < 10)**								
10 ≤ commuting time < 20	−0.570	−1.731	0.794	−0.778	0.377	0.240	1.658	−0.707
20 ≤ commuting time < 30	−1.713 **	−2.922 **	−0.646	−1.530	−1.003 *	−0.812	−1.162	−1.040
30 ≤ commuting time < 40	−4.516 ***	−4.717 **	−1.835	−7.838 ***	−1.810 **	−2.559 **	−0.235	−2.941 *
40 ≤ commuting time	−5.924 ***	−5.638 **	−5.414 *	−8.621 ***	−3.447 ***	−3.094 **	−0.169	−6.412 ***
**Commute mode (reference:** **other modes (car/motor))**								
Walking	−0.106	2.443 *	−2.711	−0.283	−0.636	−1.060	−0.838	1.888
Bicycle	1.505	1.489	1.752	2.194	0.674	−0.354	0.058	2.502 *
Public transit	0.601	2.527	−1.966	1.245	0.805	0.233	−1.403	5.693 ***
**Individual**								
Age	−0.597	0.188	−1.131	−0.968	−1.996 ***	−1.466 ***	−1.827 ***	−2.080 ***
Gender	1.916 ***	2.521 ***	1.658	1.220	−6.581 ***	−5.403 ***	−7.908 ***	−6.434 ***
Only-child	1.128	0.459	1.886	1.010	1.036 **	1.197	−0.077	1.103
Hukou	0.794	0.942	2.716 *	−2.184	2.263 ***	3.200 ***	0.656	1.835
**Family**								
Father education	0.305	0.219	−0.646	2.217	2.178 ***	2.304 ***	2.094 **	0.987
Mother education	0.177	2.344 **	−1.556	−2.906	1.526 ***	1.431 *	2.030 **	1.135
Parent expectation	2.425 ***	1.419	3.562 **	2.671 **	13.873 ***	12.765 ***	13.922 ***	14.571 ***
Family financial	2.478 ***	1.936 *	3.187 **	2.429 **	0.599	0.511	−0.060	0.327
**School**								
Teacher/student	−0.114	−0.327 *	0.154	−0.177	−0.054	0.065	−0.408 ***	−0.324 **
Teacher health	1.062	0.356	0.804	1.004	0.229	−0.564	1.504	−0.029
Teacher quality	0.005	−0.006	0.018	0.006	0.072 ***	0.072 ***	0.029	0.141 ***
School rank	−1.173 **	0.949	−2.540 **	−1.663	3.539 ***	4.074 ***	5.499 ***	0.214
Constant	72.796 ***	57.774 ***	79.706 ***	83.447 ***	72.320 ***	63.147 ***	70.499 ***	79.000 ***
Observations	4807	2135	1269	1403	4807	2135	1269	1403
R-squared	0.041	0.050	0.063	0.072	0.447	0.465	0.441	0.439

Note: * *p* < 0.1; ** *p* < 0.05; *** *p* < 0.01.

## Data Availability

The data used to support the findings of this study are available from the corresponding author upon request.

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
