# Peer review of "How School Travel Affects Children’s Psychological Well-Being and Academic Achievement in China"

_ijerph, 2022, doi:10.3390/ijerph192113881_

Round 1

Reviewer 1 Report

This study is interesting and meaningful as seldom researchers paid attention to K12 student’s commute issues in China. The result of this study found that commute time and mode were related with students’ WB and academic performance and enlightened the parents and policymakers. After reading the article, I noticed that there were some mistakes and have some suggestions—I hope those are helpful.

1.     In table 1, the author presents the description of variable Hukou and defines category “Rural=1” which I thought be “Urban=1” as the percentage in Rural Group is the smallest(0.14) and in Urban Group is the largest(0.50) according to what displayed in the table.

2.     In table4, if the reference variable of commute mode was “public transit”, the result of “car” should be present (the last row of Commute mode).

3.     As written in Introduction, the aim of the study is “ examines how commuting (time and mode) influences child’s PWB and academic performance, which will ……“(line 167)and the result also found that commute time and mode were related with PWB and AP, but the author provide more results and discussion of commute time(Section 4.4-4.5) and talked little about commute mode(only two sentences in Section4.3 and the interpretation should be reconsidered). I understand that commute mode was not a very significant variable for PWB and AP, but the result also found the differences among different commute modes, and commute time and mode might have interaction effects on PWB and AP.

Author Response

Dear Reviewer:

    We feel great thanks for your professional review work on our article “How school travel affects children’s psychological well-being and academic achievement in China” (Manuscript ID: ijerph-1953032). These comments are all valuable and helpful for improving our article. In the following, we will respond to your comments :

Point 1:: This study is interesting and meaningful as seldom researchers paid attention to K12 student’s commute issues in China. The result of this study found that commute time and mode were related with students’ WB and academic performance and enlightened the parents and policymakers. After reading the article, I noticed that there were some mistakes and have some suggestions—I hope those are helpful.

Response 1: Thank you again for your positive -pPoints and valuable suggestions to improve the quality of our manuscript.

Point 2: In table 1, the author presents the description of variable Hukou and defines category “Rural=1” which I thought be “Urban=1” as the percentage in Rural Group is the smallest (0.14) and in Urban Group is the largest (0.50) according to what displayed in the table.

Response 2: We feel sorry for our carelessness. We have corrected it in table 1(Urban =1) and we also feel great thanks for your point out.

Point 3: In table4, if the reference variable of commute mode was “public transit”, the result of “car” should be present (the last row of Commute mode).

Response 3: Thank you for pointing this out. The reference variable of commute mode was “other modes (car\motor)”, We feel sorry for our carelessness and have corrected it in table 4.

Point 4: As written in Introduction, the aim of the study is “examines how commuting (time and mode) influences child’s PWB and academic performance, which will ……“ (line 167),and the result also found that commute time and mode were related with PWB and AP, but the author provide more results and discussion of commute time(Section 4.4-4.5) and talked little about commute mode(only two sentences in Section4.3 and the interpretation should be reconsidered). I understand that commute mode was not a very significant variable for PWB and AP, but the result also found the differences among different commute modes, and commute time and mode might have interaction effects on PWB and AP.

Response 4: As the reviewer pointed out, we have added the suggested content to the manuscript.

Additionally, we discussed the relationship between transportation patterns and PWB as well as academic performance, with the following modifications:

(1)    Section4.2 -page 7- line 260-263

“Moreover, we found a correlation between commute mode and PWB. Using other modes (car/motor) as a reference group, walking to school is associated with a higher PWB (model 2) for the student in the urban centers. There are no such significant correlations in other areas.”

(2)    Section4.3 -page 8, line 279-283

“Using bicycles and public transportation increases children's academic perfor-mance in rural areas, but does not have a significant effect in urban and fringe areas. A possible explanation for this is that in China's rural areas, transportation improvement is a top priority in poverty alleviation strategies, and public transportation services have been improved [59].”

(3)    Section4.4 page9, line 298-302

“We found that the relationship between commuting mode and well-being remained consistent with the results in Table 3. On the one hand, walking to school in urban centers brings students more positive emotions compared to other commuting modes, and no such effect was found in urban fringe and rural areas. On the other hand, bicycle commuting and public transportation in rural areas had a positive effect on students' academic performance.”

(4)    Section 5 page10, line 350-358

“Third, we find that walking to school positively affects children’s PWB in urban centers. In line with other studies [57,62-65], we also found that active commuting im-proved adolescent PWB. However, this positive effect was not significant in rural areas, possibly because the environment in fringe and rural areas is not so friendly to pedestrians and cyclists, and the stress risks caused by a bad travel environment, as well as air pollution, offset the potential positive effect of active commuting [11]. Moreover, we found bicycle and public transport have a positive effect on improving the performance of rural students. Similar to the previous study by Muralidharan and Prakash (2017) in rural India [66], their study found that providing bicycles to students increased their enrollment and the likelihood that students would pass the school certificate exam. There should be adequate public service facilities in rural areas in planning management.”

We appreciate for Reviewers’ warm work, and hope that the correction will meet with approval.

Reviewer 2 Report

Thank you for the opportunity to review the manuscript " How school travel affects children’s psychological well-being (PWB) and academic achievement in China". It is generally well-written and the research presented is of empirical significance. Personally, I think it is an important topic and, although research in the area of school travel and children’s PWB and academic achievement has a long history, the current paper offers some new insights into the potential factors such as areas, commuting time, commuting mode. The literature review presents a good rationale for the study and highlights the importance of commuting on PWB and academic achievement.

The method is clearly presented. Measures used appear to be valid and reliable. Statistical analysis is appropriate. Results are clearly presented with good detail. Finally, the discussion does a good job of interpreting the findings in the light of previous research, limitations and directions for future research are noted. I wish the authors all the best in progressing this research further.

Author Response

Dear Reviewer:

    We feel great thanks for your professional review work on our article “How school travel affects children’s psychological well-being and academic achievement in China” (Manuscript ID: ijerph-1953032). These comments are all valuable and helpful for improving our article. In the following, we will respond to your comments:

Point :Thank you for the opportunity to review the manuscript " How school travel affects children’s psychological well-being (PWB) and academic achievement in China". It is generally well-written and the research presented is of empirical significance. Personally, I think it is an important topic and, although research in the area of school travel and children’s PWB and academic achievement has a long history, the current paper offers some new insights into the potential factors such as areas, commuting time, commuting mode.

The literature review presents a good rationale for the study and highlights the importance of commuting on PWB and academic achievement. The method is clearly presented. Measures used appear to be valid and reliable. Statistical analysis is appropriate. Results are clearly presented with good detail. Finally, the discussion does a good job of interpreting the findings in the light of previous research, limitations and directions for future research are noted. I wish the authors all the best in progressing this research further.

Response: Thanks for your recognition, it is very encouraging and means a lot to us, and it gives us the confidence to continue our research related to adolescent school travel and health. Our goal is to advance the transportation equity and health equity for youth through our research, and we look forward to your interest in our follow-up research.

In addition, based on your suggestions, we have added some references to our paper.

Page 10, line 349-357

“Third, we find that walking to school positively affects children’s PWB in urban centers. In line with other studies [9-14], we also found that active commuting improved adolescent mental health.”

References

Westman, J.; Johansson, M.; Olsson, L. E.; Mårtensson, F.; Friman, M. Children’S Affective Experience of Every-Day Travel. J. Transp. Geogr. 2013, 29, 95-102.

Ma, L.; Ye, R.; Wang, H. Exploring the Causal Effects of Bicycling for Transportation on Mental Health. Transportation Research Part D: Transport and Environment 2021, 93, 102773.

Sasayama, K.; Watanabe, M.; Ogawa, T. Walking to/From School is Strongly Associated with Physical Activity Before and After School and Whole-Day in Schoolchildren: A Pilot Study. J. Transp. Health 2021, 21, 101077.

Kleszczewska, D.; Mazur, J.; Bucksch, J.; Dzielska, A.; Brindley, C.; Michalska, A. Active Transport to School May Reduce Psychosomatic Symptoms in School-Aged Children: Data From Nine Countries. International Journal of Environmental Research and Public Health 2020, 17(23), 8709.

Frömel, K.; Groffik, D.; Mitáš, J.; Dygrýn, J.; Valach, P.; Aafář, M. Active Travel of Czech and Polish Adolescents in Relation to their Well-Being: Support for Physical Activity and Health. Int. J. Environ. Res. Public Health 2020, 17(6).

Phansikar, M.; Ashrafi, S. A.; Khan, N. A.; Massey, W. V.; Mullen, S. P. Active Commute in Relation to Cognition and Academic Achievement in Children and Adolescents: A Systematic Review and Future Recommendations. Int. J. Environ. Res. Public Health 2019, 16(24).

Other relevant revisions are marked in red font you can found in our revised draft, thank you again for your positive comments and valuable suggestions. We appreciate for your warm work, and hope that the correction will meet with approval.

Reviewer 3 Report

The topic of the article is interesting, especially in the context in which the research was carried out. A deficient aspect of the study is the analysis of data from the years 2013-2014 and 2014-2015. Why wasn't more recent data analyzed? The analyzed data are relevant, as a result of the large number of participants.

1) The point 2.1. it is very short. therefore, the approach to the concept of children's well-being must be developed.

2) The research questions/hypotheses are not formulated in the first part of the study.

3) In the discussion section, more correlations are needed with the data of similar research conducted in other countries.

Author Response

Dear Reviewer:

We feel great thanks for your professional review work on our article “How school travel affects children’s psychological well-being and academic achievement in China” (Manuscript ID: ijerph-1953032). These comments are all valuable and helpful for improving our article. In the following, we will respond to your comments:

Point 1: The topic of the article is interesting, especially in the context in which the research was carried out. A deficient aspect of the study is the analysis of data from the years 2013-2014 and 2014-2015. Why wasn't more recent data analyzed? The analyzed data are relevant, as a result of the large number of participants.

Response 1: We agree with the reviewer’s assessment. This paper uses data from a national tracking survey on the life, study, and health of Chinese adolescents, which can be accessed at http://ceps.ruc.edu.cn/index.htm. We used data from the 2014-2015 wave, which is the most recent year of publicly available data. In recent years, many papers have widely used this database to study adolescent health issues in China (eg.,Fang et al., 2021; Chen et al., 2017; Jia et al., 2018; Zhang et al., 2021; Yang et al., 2021; Fang et al., 2020).Many related studies that have been published in international journals using database consistent with our paper (eg. ,Liu & Jiang, 2020; Liu & Ge et al., 2020; Lu et al., 2021; Song et al., 2022; Sun et al., 2020; Wang, 2019; Zhang et al., 2020).

We will continue to update our study as new data becomes available in the future. In the meantime, we have added this to the limitations section of the paper (page 11; line 397-400):

“First, we could only obtain relevant data up to the 2014-2015 wave, and changes in school travel (commuting time and mode) may have occurred since then. We will continue to advance the research as new rounds of survey data become publicly available in the future.”

References:

Chen, R.. Li, C., Yang, A. (2017). The Impact of School Choice on Subject Well-being of Student: An Empirical Research Based on Evidence from the China Education Panel Survey. China Economic Studies, (02), 3-15. (in Chinese)

Fan, Z., Gao Y., Liu C. (2020). Nutritional Intervention,Health and Education: A Study Based on the National Nutrition Improvement Program. Finance & Trade Economics, 41(07), 21-35. (in Chinese)

Fang C., Huang B. (2021). Can Physical Exercise Promote the Development of Teenagers’ Cognitive Ability? An Empirical Study Based on CEPS. JOURNAL OF EAST CHINA NORMAL UNIVERSITY Educational Sciences, 39(03), 84-98. (in Chinese)

Liu, Y., Ge, T., & Jiang, Q. (2020). Changing family relationships and mental health of Chinese adolescents: the role of living arrangements. PUBLIC HEALTH, 186, 110-115.

Liu, Y., & Jiang, Q. (2020). Who Benefits From Being an Only Child? A Study of Parent-Child Relationship Among Chinese Junior High School Students. Frontiers in Psychology, 11, 608995.

Lu, H., Nie, P., & Sousa-Poza, A. (2021). The effect of parental educational expectations on adolescent subjective well-being and the moderating role of perceived academic pressure: longitudinal evidence for China. Child Indicators Research, 14(1), 117-137.

Jin, J., Li, X., Wang, H. (2018). Win at the Starting Line? An Empirical Study on Preschool Education Experience and the Development of Multiple Abilities of Adolescents . EDUCATION&ECONOMY (06), 56-64. (in Chinese)

Slobodin, O., & Davidovitch, M. (2022). Primary School Children's Self-Reports of Attention Deficit Hyperactivity Disorder-Related Symptoms and Their Associations With Subjective and Objective  Measures of Attention Deficit Hyperactivity Disorder [Journal Article]. Frontiers in Human Neuroscience, 16, 806047.

Song, Y. P., Hu, X. C., & Long, X. (2022). [Influence of free nutrition lunch policy on overweight of rural adolescents: Based on China Education Panel Survey Junior High School Cohort data] Chinese Journal of Epidemiolog , 43(6), 885-891.

Stine, F., Collier, D. N., Fang, X., Dew, K. R., & Lazorick, S. (2021). Impact of Body Mass Index, Socioeconomic Status, and Bedtime Technology Use on Sleep Duration in Adolescents CLINICAL PEDIATRICS, 60(13), 520-527.

Sun, L., Shafiq, M. N., McClure, M., & Guo, S. (2020). Are there educational and psychological benefits from private supplementary tutoring in Mainland China? Evidence from the China Education Panel Survey, 2013–15. INTERNATIONAL JOURNAL OF EDUCATIONAL DEVELOPMENT, 72, 102144.

Wang, D. (2019). Reduction but not elimination: health inequalities among urban, migrant, and rural children in China-the moderating effect of the fathers' education level. BMC PUBLIC HEALTH, 19(1), 1219.

Yang, L., Dai Y. (2021). Family Structure and Adolescent Health: Mediating Effects of Parental Involvement and Non-Cognitive Abilities. Social Construction, 8(05), 55-72. (in Chinese)

Zhang, L., Wang, W., Dong, X., Zhao, L., Peng, J., & Wang, R. (2020). Association between time spent outdoors and myopia among junior high school students. MEDICINE, 99(50), e23462.

Zhang, L., Lu, C., Chen, B. (2021). Pathways by which physical activity affects secondary school students' academic performance. Youth Studies (06), 70-82. (in Chinese)

Point 2: The point 2.1. it is very short. therefore, the approach to the concept of children's well-being must be developed.

Response 2: As suggested by the referee, we have added the suggested content to the manuscript on page 3 line 100-117:

“Well-being (WB) is a term that is commonly used but difficult to define [31]. A number of studies have been conducted on WB from two different perspectives. On the one hand, according to the Hedonic perspective, WB is related to happiness, life satis-faction, and quality of life [32,33]. On the other hand, the eudaimonic perspective focuses on people's self-actualization and personal growth [34,35]. No matter which perspective is adopted, there seems to be a consensus that WB is a multidimensional indicator [31,36,37].

Similarly, children's well-being, measuring the condition of their lives, cannot also be represented by a single indicator [38,39]. According to Pollard and Lee (2003), children's well-being can be defined as a multidimensional concept across five domains: physical, psychological, social, cognitive, and economic [37]. Furthermore, they distinguished between psychological well-being and cognitive well-being. The term psychological well-being refers to indicators of emotional well-being, mental health, and illness, while cognitive well-being refers to indicators of intelligence and school performance (such as academic performance). A recent integrative review by Waygood et al. (2017) examines how transport impacts on children's five domains of WB, two of which relate to psychological and cognitive domains [17,40]. In the following sections, we will discuss these two types of WB in greater detail.”

Point 3: The research questions/hypotheses are not formulated in the first part of the study.

Response3: As suggested by the referee, we have added the suggested content to the manuscript on page 2 line 87-94:

“The aim of this study is to examine the relationships between commuting, children's psychological well-being, and academic performance using the Chinese Education Panel Survey (CEPS2014-2015). In order to fill the aforementioned knowledge gaps, this study identifies: (1) How commuting (time/mode) influences a child's psychological well-being. (2) How commuting (time/mode) influences child’s academic performance. (3) Whether commute time, child's psychological well-being, and academic performance differ across different groups (considering urban and rural areas, psychological WB and academic performance distribution).”

Point 4: In the discussion section, more correlations are needed with the data of similar research conducted in other countries.

Response 4: As the reviewer pointed out, we rewrite the discussion , have added more correlations and new citations to this section:

  • Page 10, line 334-345

“First, we find that long commuting time is associated with lower PWB in China. It is consistent with results from Sweden children [19], which found that a short journey results in a more positive mood. Janáček(2020) also observed that Czech students with long commute times reported a lower level of life satisfaction, indicating that commute time plays an important role in improving student well-being [47]. Looking at differ-ences by urban-rural area, students living in urban centers are more sensitive to longer commute times than other areas. One reason may be that with rapidly increasing motorization in China, and traffic congestion is getting more serious. People living in the city centers face more traffic and noise, and those traffic-induced noise pollution may attenuate the residents’ mental health [60]. According to a recent national study of mental health, with nearly 30 million Chinese adolescents under 17 years old suffering from mental anxiety requiring comprehensive interventions[61], our study prove that reducing commuting time is an effective intervention to enhance Chinese adolescents’ PWB.”

  • Page 10, line 346-349

“Second, we find that commute time negatively affects students' academic performance. This finding in line with the evidence from Brazil and Chile [54,55], with children who traveled longer performed significantly worse than children with shorter school journeys”

  • Page 10, line 361-371

“Third, we find that walking to school positively affects children’s PWB in urban centers. In line with other studies [57,62-65], we also found that active commuting im-proved adolescent PWB. However, this positive effect was not significant in rural areas, possibly because the environment in fringe and rural areas is not so friendly to pedestrians and cyclists, and the stress risks caused by a bad travel environment, as well as air pollution, offset the potential positive effect of active commuting [11]. Moreover, we found bicycle and public transport have a positive effect on improving the performance of rural students. Similar to the previous study by Muralidharan and Prakash (2017) in rural India [66], the study finds that providing bicycles to students increased their enrollment and the likelihood that students would pass the school certificate exam. There should be adequate public service facilities in rural areas in planning management.”

We appreciate for Editors’/Reviewers’ warm work, and hope that the correction will meet with approval.”

References:

  1. Dodge, R.; Daly, A. P.; Huyton, J. L.; Sanders, L. The Challenge of Defining Wellbeing. International Journal of Wellbeing 2012, 2, 222-235.
  2. Diener; Ed Subjective Well-Being. Psychol. Bull. 1984, 95(3), 542-575.
  3. Diener; Edsuh; Lucas, E. M.; Smith, R. E.; Heidi, L. Subjective Well-Being: Three Decades of Progress. Psychol. Bull. 1999.
  4. Waterman, A. S. Two Conceptions of Happiness: Contrasts of Personal Expressiveness (Eudaimonia) and Hedonic Enjoyment. J. Pers. Soc. Psychol. 1993, 64(4), 678.
  5. Ryff, C. D. Happiness is Everything, Or is It? Explorations On the Meaning of Psychological Well-Being. J. Pers. Soc. Psychol. 1989, 57(6), 1069.
  6. Lee, P. P. C. D. Child Well-Being: A Systematic Review of the Literature. Soc. Indic. Res. 2003, 61(1), 59-78.
  7. Diener, E.; Oishi, S.; Tay, L. Advances in Subjective Well-Being Research. Nat. Hum. Behav. 2018, 2(4), 253-260.
  8. Westman, J.; Johansson, M.; Olsson, L. E.; Mårtensson, F.; Friman, M. Children’S Affective Experience of Every-Day Travel. J. Transp. Geogr. 2013, 29, 95-102.
  9. Ma, L.; Ye, R.; Wang, H. Exploring the Causal Effects of Bicycling for Transportation On Mental Health. Transportation Research Part D: Transport and Environment 2021, 93, 102773.
  10. Sasayama, K.; Watanabe, M.; Ogawa, T. Walking to/From School is Strongly Associated with Physical Activity Before and After School and Whole-Day in Schoolchildren: A Pilot Study. J. Transp. Health 2021, 21, 101077.
  11. Kleszczewska, D.; Mazur, J.; Bucksch, J.; Dzielska, A.; Brindley, C.; Michalska, A. Active Transport to School May Reduce Psychosomatic Symptoms in School-Aged Children: Data From Nine Countries. International Journal of Environmental Research and Public Health 2020, 17(23), 8709.
  12. Frömel, K.; Groffik, D.; Mitáš, J.; Dygrýn, J.; Valach, P.; Aafář, M. Active Travel of Czech and Polish Adolescents in Relation to their Well-Being: Support for Physical Activity and Health. Int. J. Environ. Res. Public Health 2020, 17(6).
  13. Phansikar, M.; Ashrafi, S. A.; Khan, N. A.; Massey, W. V.; Mullen, S. P. Active Commute in Relation to Cognition and Academic Achievement in Children and Adolescents: A Systematic Review and Future Recommendations. Int. J. Environ. Res. Public Health 2019, 16(24).
  14. Stark, J.; Meschik, M.; Singleton, P. A.; Schützhofer, B. Active School Travel, Attitudes and Psychological Well-Being of Children. Transportation Research Part F: Traffic Psychology and Behaviour 2018, 56, 453-465.

Round 2

Reviewer 3 Report

The authors reviewed all the mentioned aspects. At this moment, the manuscript meets all the criteria to be published.